# Learning to Coordinate Manipulation Skills via Skill Behavior Diversification

**Youngwoon Lee, Jingyun Yang, Joseph J. Lim**
Department of Computer Science
University of Southern California
{lee504,jingyuny,limjj}@usc.edu

## Abstract

When mastering a complex manipulation task, humans often decompose the task into sub-skills of their body parts, practice the sub-skills independently, and then execute the sub-skills together. Similarly, a robot with multiple end-effectors can perform complex tasks by coordinating sub-skills of each end-effector. To realize temporal and behavioral coordination of skills, we propose a modular framework that first individually trains sub-skills of each end-effector with skill behavior diversification, and then learns to coordinate end-effectors using diverse behaviors of the skills. We demonstrate that our proposed framework is able to efficiently coordinate skills to solve challenging collaborative control tasks such as picking up a long bar, placing a block inside a container while pushing the container with two robot arms, and pushing a box with two ant agents. Videos and code are available at https://clvrai.com/coordination.

## 1 Introduction

Imagine you wish to play Chopin's Fantaisie Impromptu on the piano. With little prior knowledge about the piece, you would first practice playing the piece with each hand separately. After independently mastering the left and right hand parts, you would move on to practicing with both hands simultaneously. To find the synchronized and non-interfering movements of two hands, you would try variable ways of playing the same melody with each hand, and eventually create a complete piece of music. Through the decomposition of skills into sub-skills of two hands and learning variations of sub-skills, humans make the learning process of manipulation skills much faster than learning everything at once.

Can autonomous agents efficiently learn complicated tasks with coordination of different skills from multiple end-effectors like humans? Learning to perform collaborative and composite tasks from scratch requires a huge amount of environment interaction and extensive reward engineering, which often results in undesired behaviors (Riedmiller et al., 2018). Hence, instead of learning a task at once, modular approaches (Andreas et al., 2017; Oh et al., 2017; Frans et al., 2018; Lee et al., 2019; Peng et al., 2019; Goyal et al., 2020) suggest to learn reusable primitive skills and solve more complex tasks by recombining the skills. However, all these approaches either focus on working with single end-effector manipulation or single agent locomotion, and these do not scale to multi-agent problems.

To this end, we propose a modular framework that learns to coordinate multiple end-effectors with their primitive skills for various robotics tasks, such as bimanual manipulation. The main challenge is that naive simultaneous execution of primitive skills from multiple end-effectors can often cause unintended behaviors (e.g. collisions between end-effectors). Thus, as illustrated in Figure 1, our model needs to learn to appropriately coordinate end-effectors; and hence needs a way to obtain, represent, and control detailed behaviors of each primitive skill. Inspired by these intuitions, our method consists of two parts: (1) acquiring primitive skills with diverse behaviors by mutual information maximization, and (2) learning a meta policy that selects a skill for each end-effector and coordinates the chosen skills by controlling the behavior of each skill.

The main contribution of this paper is a modular and hierarchical approach that tackles cooperative manipulation tasks with multiple end-effectors by (1) learning primitive skills of each end-effector independently with skill behavior diversification and (2) coordinating end-effectors using diverse

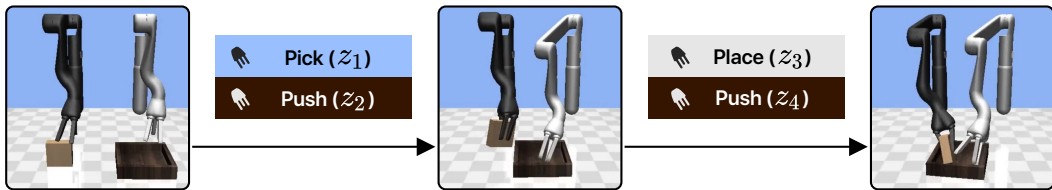

Figure 1: Composing complex skills using multiple agents' primitive skills requires proper coordination between agents since concurrent execution of primitive skills requires temporal and behavioral coordination. For example, to move a block into a container on the other end of the table, the agent needs to not only utilize pick, place, and push primitive skills at the right time but also select the appropriate behaviors for these skills, represented as latent vectors $z_1$, $z_2$, $z_3$, and $z_4$ above. Naive methods neglecting either temporal or behavioral coordination will produce unintended behaviors, such as collisions between end-effectors.

behaviors of the skills. Our empirical results indicate that our proposed method is able to efficiently learn primitive skills with diverse behaviors and coordinate these skills to solve challenging collaborative control tasks such as picking up a long bar, placing a block inside the container on the right side, and pushing a box with two ant agents. We provide additional qualitative results and code at `https://clvrai.com/coordination`.

## 2 RELATED WORK

Deep reinforcement learning (RL) for continuous control is an active research area. However, learning a complex task either from a sparse reward or a heavily engineered reward becomes computationally impractical as the target task becomes complicated. Instead of learning from scratch, complex tasks can be tackled by decomposing the tasks into easier and reusable sub-tasks. Hierarchical reinforcement learning temporally splits a task into a sequence of temporally extended meta actions. It often consists of one meta policy (high-level policy) and a set of low-level policies, such as options framework (Sutton et al., 1999). The meta policy decides which low-level policy to activate and the chosen low-level policy generates an action sequence until the meta policy switches it to another low-level policy. Options can be discovered without supervision (Schmidhuber, 1990; Bacon et al., 2017; Nachum et al., 2018; Levy et al., 2019), meta-learned (Frans et al., 2018), pre-defined (Kulkarni et al., 2016; Oh et al., 2017; Merel et al., 2019; Lee et al., 2019), or attained from additional supervision signals (Andreas et al., 2017; Ghosh et al., 2018). However, option frameworks are not flexible to solve a task that requires simultaneous activation or interpolation of multiple skills since only one skill can be activated at each time step.

To solve composite tasks multiple policies can be simultaneously activated by adding Q-functions (Haarnoja et al., 2018a), additive composition (Qureshi et al., 2020; Goyal et al., 2020), or multiplicative composition (Peng et al., 2019). As each policy takes the whole observation as input and controls the whole agent, it is not robust to changes in unrelated parts of the observation. For example, a left arm skill can be affected by the pose change in the right arm, which is not relevant to the left arm skill. Hence, these skill composition approaches can fail when an agent encounters a new combination of skills or a new skill is introduced since the agent will experience unseen observations.

Instead of having a policy with the full observation and action space, multi-agent reinforcement learning (MARL) suggests to explicitly split the observation and action space according to agents (e.g. robots or end-effectors), which allows efficient low-level policy training as well as flexible skill composition. For cooperative tasks, communication mechanisms (Sukhbaatar et al., 2016; Peng et al., 2017; Jiang & Lu, 2018), sharing policy parameters (Gupta et al., 2017), and decentralized actor with centralized critic (Lowe et al., 2017; Foerster et al., 2018) have been actively used. However, these approaches suffer from the credit assignment problem (Sutton, 1984) among agents and the lazy agent problem (Sunehag et al., 2018). As agents have more complicated morphology and larger observation space, learning a policy for a multi-agent system from scratch requires extremely long training time. Moreover, the credit assignment problem becomes more challenging when the complexity of cooperative tasks increases and all agents need to learn completely from scratch. To resolve these issues, we propose to first train reusable skills for each agent in isolation, instead of

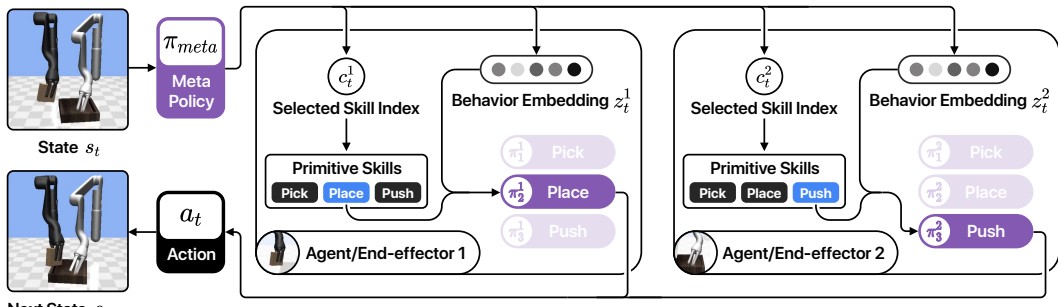

Figure 2: Our method is composed of two components: a meta policy and a set of agent-specific primitive policies relevant to task completion. The meta policy selects which primitive skill to run for each agent as well as the behavior embedding (i.e. variation in behavior) of the chosen primitive skill. Each selected primitive skill takes as input the agent observation and the behavior embedding and outputs action for that agent.

learning primitive skills of multiple agents together. Then, we recombine these skills (Maes & Brooks, 1990) to complete more complicated tasks with learned coordination of the skills.

To coordinate skills from multiple agents, the skills have to be flexible; hence, a skill can be adjusted to collaborate with other agents' skills. Maximum entropy policies (Haarnoja et al., 2017; 2018a;b) can learn diverse ways to achieve a goal by maximizing not only reward but also entropy of the policy. In addition, Eysenbach et al. (2019) proposes to discover diverse skills without reward by maximizing entropy as well as mutual information between resulting states and latent representations of skills (i.e. skill embeddings). Our method leverages the maximum entropy policy (Haarnoja et al., 2018b) with the discriminability objective (Eysenbach et al., 2019) to learn a primitive skill with diverse behaviors conditioned on a controllable skill embedding. This controllable skill embedding will be later used as a *behavior embedding* for the meta policy to adjust a primitive skill's behavior for coordination.

## 3 METHOD

In this paper, we address the problem of solving cooperative manipulation tasks that require collaboration between multiple end-effectors or agents. Note that we use the terms "end-effector" and "agent" interchangeably in this paper. Instead of learning a multi-agent task from scratch (Lowe et al., 2017; Gupta et al., 2017; Sunehag et al., 2018; Foerster et al., 2018), modular approaches (Andreas et al., 2017; Frans et al., 2018; Peng et al., 2019) suggest to learn reusable primitive skills and solve more complex tasks by recombining these skills. However, concurrent execution of primitive skills of multiple agents fails when agents never experienced a combination of skills during the pre-training stage, or skills require temporal or behavioral coordination.

Therefore, we propose a modular and hierarchical framework that learns to coordinate multiple agents with primitive skills to perform a complex task. Moreover, during primitive skill training, we propose to learn a latent *behavior embedding*, which provides controllability of each primitive skill to the meta policy while coordinating skills. In Section 3.2, we describe our modular framework in detail. Next, in Section 3.3, we elaborate how controllable primitive skills can be acquired. Lastly, we describe how the meta policy learns to coordinate primitive skills in Section 3.4.

### 3.1 PRELIMINARIES

We formulate our problem as a Markov decision process defined by a tuple $\{\mathcal{S}, \mathcal{A}, \mathcal{T}, R, \rho, \gamma\}$ of states, actions, transition probability, reward, initial state distribution, and discount factor. In our formulation, we assume the environment includes $N$ agents. To promote consistency in our terminology, we use superscripts to denote the index of agent and subscripts to denote time or primitive skill index. Hence, the state space and action space for an agent $i$ can be represented as $\mathcal{S}^i$ and $\mathcal{A}^i$ where each element of $\mathcal{S}^i$ is a subset of the corresponding element in $\mathcal{S}$ and $\mathcal{A} = \mathcal{A}^1 \times \mathcal{A}^2 \times \cdots \times \mathcal{A}^N$, respectively. For each agent $i$, we provide a set of $m^i$ skills, $\Pi^i = \{\pi_1^i, \ldots, \pi_{m^i}^i\}$. A policy of an agent $i$ is represented as

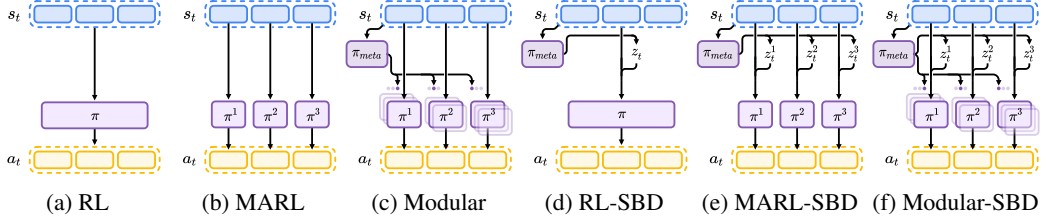

(a) RL     (b) MARL     (c) Modular     (d) RL-SBD     (e) MARL-SBD     (f) Modular-SBD

Figure 3: Different multi-agent architectures. (a) The vanilla RL method considers all agents as a monolithic agent; thus a single policy takes the full observation as input and outputs the full action. (b) The multi-agent RL method (MARL) consists of $N$ policies that operate on the observations and actions of corresponding agents. (c) The modular network consists of $N$ sets of skills for the $N$ agents trained in isolation and a meta policy that selects a skill for each agent. (d-f) The RL, MARL, and modular network methods augmented with skill behavior diversification (SBD) has a meta policy that outputs a skill behavior embedding vector $z$ for each skill.

$\pi_{c_t^i}^i(a_t^i|s_t^i) \in \Pi^i$, where $c_t^i$ is a skill index, $s_t^i \in \mathcal{S}^i$ is a state, and $a_t^i \in \mathcal{A}^i$ is an agent action at time $t$. An initial state $s_0$ is sampled from $\rho$, and then, $N$ agents take actions $a_t^1, a_t^2, \ldots, a_t^N$ sampled from a composite policy $\pi(a_t^1, a_t^2, \ldots, a_t^N|s_t, c_t^1, c_t^2, \ldots, c_t^N) = (\pi_{c_t^1}^1(a_t^1|s_t), \pi_{c_t^2}^2(a_t^2|s_t), \ldots, \pi_{c_t^N}^N(a_t^N|s_t))$ and receive a single reward $r_t$. The performance is evaluated based on a discounted return $R = \sum_{t=0}^{T-1} \gamma^t r_t$, where $T$ is the episode horizon.

## 3.2 MODULAR FRAMEWORK

As illustrated in Figure 2, our model is composed of two components: a meta policy $\pi_{meta}$ and a set of primitive skills of $N$ agents $\Pi^1, \ldots, \Pi^N$. Note that each primitive skill $\pi_{c^i}^i \in \Pi^i$ contains variants of behaviors parameterized by an $N_z$-dimensional latent behavior embedding $z^i$ (see Section 3.3). The meta policy selects a skill to execute for each agent, rather than selecting one primitive skill for the entire multi-agent system to execute. Also, we give the meta policy the capability to select which variant of the skill to execute (see Section 3.4). Then, the chosen primitive skills are simultaneously executed for $T_{low}$ time steps.

The concurrent execution of multiple skills often leads to undesired results and therefore requires coordination between the skills. For example, naively placing a block in the left hand to a container being moved by the right hand can cause collision between the two robot arms. The arms can avoid collision while performing the skills by properly adjusting their skill behaviors (e.g. the left arm leaning to the left side while placing the block and the right arm leaning to the right side while pushing the container) as shown in Figure 1. In our method, the meta policy learns to coordinate multiple agents' skills by manipulating the behavior embeddings (i.e. selecting a proper behavior from diverse behaviors of each skill).

## 3.3 TRAINING AGENT-SPECIFIC PRIMITIVE SKILLS WITH DIVERSE BEHAVIORS

To adjust a primitive skill to collaborate with other agents' skills in a new environment, the skill needs to support variations of skill behaviors when executed at a given state. Moreover, a behavioral variation of a skill should be controllable by the meta policy for skill coordination. In order to make our primitive skill policies generate diverse behaviors controlled by a latent vector $z$, we leverage the entropy and mutual information maximization objective introduced in Eysenbach et al. (2019).

More specifically, a primitive policy of an agent $i$ outputs an action $a \in \mathcal{A}$ conditioned on the current state $s \in \mathcal{S}$ and a latent behavior embedding $z \sim p(z)$, where the prior distribution $p(z)$ is Gaussian (we omit agent $i$ in this section for the simplicity of notations). Diverse behaviors conditioned on a random sample $z$ can be achieved by maximizing the mutual information between behaviors and states $MI(s, z)$, while minimizing the mutual information between behaviors and actions given the state $MI(a, z|s)$, together with maximizing the entropy of the policy $\mathcal{H}(a|s)$ to encourage diverse behaviors. The objective can be written as follows (we refer the readers to Eysenbach et al. (2019)

---

**Algorithm 1** ROLLOUT

---

1: **Input:** Meta policy $\pi_{meta}$, sets of primitive policies $\Pi^1, ..., \Pi^N$, and meta horizon $T_{low}$
2: Initialize an episode $t \leftarrow 0$ and receive initial state $s_0$
3: **while** episode is not terminated **do**
4:     Sample skill indexes and behavior embeddings $(c_t^1, \ldots, c_t^N), (z_t^1, \ldots, z_t^N) \sim \pi_{meta}(s_t)$
5:     $\tau \leftarrow 0$
6:     **while** $\tau < T_{low}$ and episode is not terminated **do**
7:         $a_{t+\tau} = (a_{t+\tau}^1, \ldots, a_{t+\tau}^N) \sim (\pi_{c_t^1}^1(s_{t+\tau}, z_t^1), \ldots, \pi_{c_t^N}^N(s_{t+\tau}, z_t^N))$
8:         $s_{t+\tau+1}, r_{t+\tau} \leftarrow \text{ENV}(s_{t+\tau}, a_{t+\tau})$
9:         $\tau \leftarrow \tau + 1$
10:     **end while**
11:     Add a transition $s_t, (c_t^1, \ldots, c_t^N), (z_t^1, \ldots, z_t^N), s_{t+\tau}, r_{t:t+\tau-1}$ to the rollout buffer $\mathcal{B}$
12:     $t \leftarrow t + \tau$
13: **end while**

---

for derivation):

$$\mathcal{F}(\theta) = MI(s,z) - MI(a,z|s) + \mathcal{H}(a|s) = \mathcal{H}(a|s,z) - \mathcal{H}(z|s) + \mathcal{H}(z) \tag{1}$$

$$= \mathcal{H}(a|s,z) + \mathbb{E}_{z \sim p(z), s \sim \pi(z)}[\log p(z|s)] - \mathbb{E}_{z \sim p(z)}[\log p(z)] \tag{2}$$

$$\geq \mathcal{H}(a|s,z) + \mathbb{E}_{z \sim p(z), s \sim \pi(z)}[\log q_\phi(z|s) - \log p(z)], \tag{3}$$

where the learned discriminator $q_\phi(z|s)$ approximates the posterior $p(z|s)$.

To achieve a primitive skill with diverse behaviors, we augment Equation (3) to the environment reward:

$$r_t + \lambda_1 \mathcal{H}(a|s,z) + \lambda_2 \mathbb{E}_{z \sim p(z), s \sim \pi(z)}[\log q_\phi(z|s) - \log p(z)], \tag{4}$$

where $\lambda_1$ is the entropy coefficient and $\lambda_2$ is the diversity coefficient which corresponds identifiability of behaviors. Maximizing Equation (3) encourages multi-modal exploration strategies while maximizing the reward $r_t$ forces to achieve its own goal. Moreover, by maximizing identifiability of behaviors, the latent vector $z$, named *behavior embedding*, can represent a variation of the learned policy and thus can be used to control the behavior of the policy. For example, when training a robot to move an object, a policy learns to move the object quickly as well as slowly, and these diverse behaviors map to different latent vectors $z$. We empirically show that the policies with diverse behaviors achieve better compositionality with other agents in our experiments.

### 3.4 COMPOSING PRIMITIVE SKILLS WITH META POLICY

We denote the meta policy as $\pi_{meta}(c^1, \ldots, c^N, z^1, \ldots, z^N|s_t)$, where $c^i \in [1, m^i]$ represents a skill index of an agent $i \in [1, N]$ and $z^i \in \mathbb{R}^{N_z}$ represents a behavior embedding of the skill. Every $T_{low}$ time steps, the meta policy chooses one primitive skill $\pi_{c^i}^i \in \Pi^i$ for each agent $i$. Also, the meta policy outputs a set of latent behavior embeddings $(z^1, z^2, \ldots, z^N)$ and feeds them to the corresponding skills (i.e. $\pi_{c^i}^i(a^i|s^i, z^i)$ for agent $i$). Once a set of primitive skills $\{\pi_{c^1}^1, \ldots, \pi_{c^N}^N\}$ are chosen to be executed, each primitive skill generates an action $a^i \sim \pi_{c^i}^i(a^i|s^i, z^i)$ based on the current state $s^i$ and the latent vector $z^i$. Algorithm 1 illustrates the overall rollout process.

Since there are a finite number of skills for each agent to execute, the meta action space for each agent $[1, m^i]$ is discrete, while the behavior embedding space for each agent $\mathbb{R}^{N_z}$ is continuous. Thus, the meta policy is modeled as a $(2 \times N)$-head neural network where the first $N$ heads represent $m^i$-way categorical distributions for skill selection and the last $N$ heads represent $N_z$-dimensional Gaussian distributions for behavior control of the chosen skill.

### 3.5 IMPLEMENTATION

We model the primitive policies and posterior distributions $q_\phi$ as neural networks. We train the primitive policies using soft actor-critic (Haarnoja et al., 2018b). When we train a primitive policy, we use a unit Gaussian distribution as the prior distribution of latent variables $p(z)$. We use 5 as the size of latent behavior embedding $N_z$. Each primitive policy outputs the mean and standard

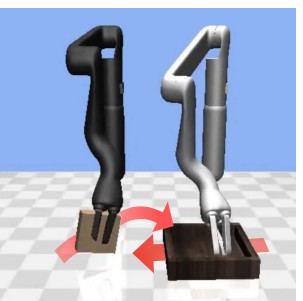 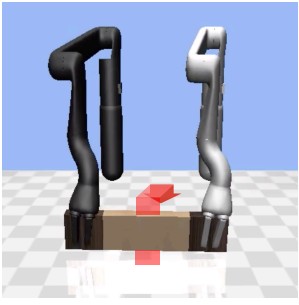 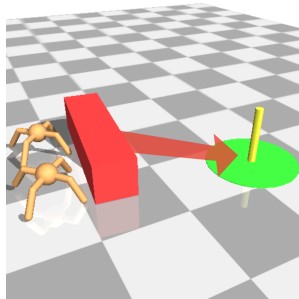

(a) JACO PICK-PUSH-PLACE     (b) JACO BAR-MOVING     (c) ANT PUSH

Figure 4: The composite tasks pose a challenging combination of object manipulation and locomotion skills, which requires coordination of multiple agents and temporally extended behaviors. (a) The left Jaco arm needs to pick up a block while the right Jaco arm pushes a container, and then it places the block into the container. (b) Two Jaco arms are required to pick and place a bar-shaped block together. (c) Two ants push the red box to the goal location (green circle) together.

deviation of a Gaussian distribution over an action space. For a primitive policy, we apply $\tanh$ activation to normalize the action between $[-1, 1]$. We model the meta policy as neural network with multiple heads that output the skill index $c^i$ and behavior embedding $z^i$ for each agent. The meta policy is trained using PPO (Schulman et al., 2017; 2016; Dhariwal et al., 2017). All policy networks in this paper consist of 3 fully connected layers of 64 hidden units with ReLU nonlinearities. The discriminator $q_\phi$ in Equation (4) is a 2-layer fully connected network with 64 hidden units.

## 4 EXPERIMENTS

To demonstrate the effectiveness of our framework, we compare our method to prior methods in the field of multi-agent RL and ablate the components of our framework to understand their importance. We conducted experiments on a set of challenging robot control environments that require coordination of different agents to complete collaborative robotic manipulation and locomotion tasks.

Through our experiments, we aim to answer the following questions: (1) can our framework efficiently learn to combine primitive skills to execute a complicated task; (2) can our learned agent exhibit collaborative behaviors during task execution; and (3) can our framework leverage the controllable behavior variations of the primitive skills to achieve better coordination?

For details about environments and training, please refer to the supplementary material. As the performance of training algorithms varies between runs, we train each method on each task with 6 different random seeds and report mean and standard deviation of each method's success rate.

### 4.1 BASELINES

We compare the performance of our method with various single- and multi-agent RL methods illustrated in Figure 3:

**Single-agent RL (RL):** A vanilla RL method where a single policy takes as input the full observation and outputs all agents' actions.

**Multi-agent RL (MARL):** A multi-agent RL method where each of $N$ policies takes as input the observation of the corresponding agent and outputs an action for that agent. All policies share the global critic learned from a single task reward (Lowe et al., 2017).

**Modular Framework (Modular):** A modular framework composed of a meta policy and $N$ sets of primitive skills (i.e. one or more primitive skills per agent). Every $T_{low}$ time steps, the meta policy selects a primitive skill for each agent based on the full observation. Then, the chosen skills are executed for $T_{low}$ time steps.

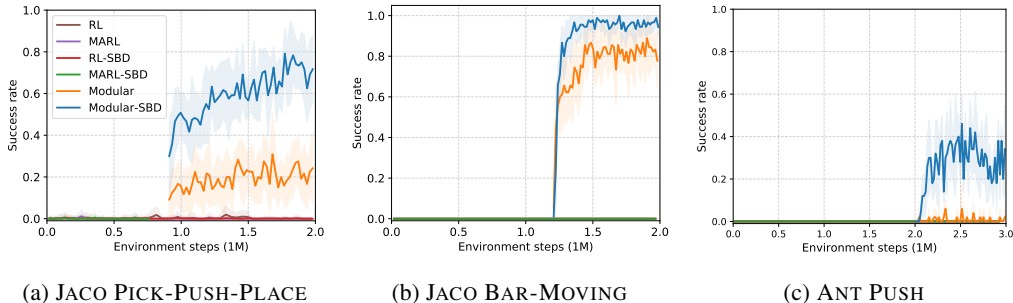

(a) JACO PICK-PUSH-PLACE            (b) JACO BAR-MOVING            (c) ANT PUSH

Figure 5: Success rates of our method (Modular-SBD) and baselines. For modular frameworks (Modular and Modular-SBD), we shift the learning curves rightwards the total number of environment steps the agent takes to learn the primitive skills (0.9 M, 1.2 M, and 2.0 M, respectively). Our method substantially improves learning speed and performance on JACO PICK-PUSH-PLACE and ANT PUSH. The shaded areas represent the standard deviation of results from six different seeds. The curves are smoothed using moving average over 10 runs.

**Single-agent RL with Skill Behavior Diversification (RL-SBD):** An **RL** method augmented with the behavior diversification objective. A meta policy is employed to generate a behavior embedding for a low-level policy, and the low-level policy outputs all agents' actions conditioned on the behavior embedding and the full observation for $T_{low}$ time steps. The meta policy and the low-level policy are jointly trained with the behavior diversification objective described in Equation (4).

**Multi-agent RL with Skill Behavior Diversification (MARL-SBD):** A **MARL** method augmented with the behavior diversification objective. A meta policy generates $N$ behavior embeddings. Then, each low-level policy outputs each agent's action conditioned on its observation and behavior embedding for $T_{low}$ time steps. All policies are jointly trained to maximize Equation (4).

**Modular Framework with Skill Behavior Diversification (Modular-SBD, Ours):** Our method which coordinates primitive skills of multiple agents. The modular framework consists of a meta policy and $N$ sets of primitive skills, where each primitive skill is conditioned on a behavior embedding $z$. The meta policy takes as input the full observation and selects both a primitive skill and a behavior embedding for each agent. Then, each primitive skill outputs action for each agent.

## 4.2    JACO PICK-PUSH-PLACE

We developed JACO PICK-PUSH-PLACE and JACO BAR-MOVING environments using two Kinova Jaco arms, where each Jaco arm is a 9 DoF robotic arm with 3 fingers. JACO PICK-PUSH-PLACE starts with a block on the left and a container on the right. The robotic arms need to pick up the block, push the container to the center, and place the block inside the container. For successful completion of the task, the two Jaco arms have to concurrently execute their distinct sets of skills and dynamically adjust their picking, pushing, and placing directions to avoid collision between arms.

**Primitives skills.** There are three primitive skills available to each arm: *Picking up*, *Pushing*, and *Placing to center* (see Figure 4a). *Picking up* requires a robotic arm to pick up a small block, which is randomly placed on the table. If the block is not picked up after a certain amount of time or the arm drops the block, the agent fails. *Pushing* learns to push a big container to its opposite side (e.g. from left to the center or from right to center). The agent fails if it cannot place the container to the center. *Placing to center* requires placing an object in the gripper to the table. The agent only succeeds when it stably places the object at the desired location on the container.

**Composite task.** Our method (Modular-SBD) can successfully perform JACO PICK-PUSH-PLACE task while all baselines fail to compose primitive skills as shown in Figure 5a. The RL and MARL baselines cannot learn the composite task mainly because the agent requires to learn the combinatorial number of skill compositions and to solve the credit assignment problem across multiple agents. Since the composite task requires multiple primitive skills of multiple agents to be performed properly at the same time, a reward signal about a failure case cannot be assigned to the correct agent or skill. By using pre-trained primitive skills, the credit assignment problem is relaxed and all agents can

|  | Jaco Pick-Push-Place | Jaco Bar-Moving | Ant Push |
|---|---|---|---|
| RL | $0.000 \pm 0.000$ | $0.000 \pm 0.000$ | $0.000 \pm 0.000$ |
| MARL | $0.000 \pm 0.000$ | $0.000 \pm 0.000$ | $0.000 \pm 0.000$ |
| RL-SBD | $0.000 \pm 0.000$ | $0.000 \pm 0.000$ | $0.000 \pm 0.000$ |
| MARL-SBD | $0.000 \pm 0.000$ | $0.000 \pm 0.000$ | $0.000 \pm 0.000$ |
| Modular | $0.324 \pm 0.468$ | $0.917 \pm 0.276$ | $0.003 \pm 0.058$ |
| Modular-SBD (Ours) | $\mathbf{0.902 \pm 0.298}$ | $\mathbf{0.950 \pm 0.218}$ | $\mathbf{0.323 \pm 0.468}$ |

Table 1: Success rates for all tasks, comparing our method against baselines. Each entry in the table represents average success rate and standard deviation over 100 runs. The baselines learning from scratch fail to learn complex tasks with multiple agents.

perform their skills concurrently. Therefore, the Modular baseline learns to achieve success but shows significantly lower performance than our method (Modular-SBD). This is because the lack of skill behavior diversification makes it impossible to adjust pushing and placing trajectories during skill composition time, which resulting in frequent end-effector collisions.

### 4.3 JACO BAR-MOVING

In JACO BAR-MOVING, two Jaco arms need to pick up a long bar together, move the bar towards a target location while maintaining its rotation, and place it on the table (see Figure 4b). The initial position of the bar is randomly initialized every episode and an agent needs to find appropriate coordination between two arms for each initialization. Compared to JACO PICK-PUSH-PLACE, this task requires that the two arms synchronize their movements and perform more micro-level adjustments to their behaviors.

**Primitives skills.** There are two pre-trained primitive skills available to each arm: *Picking up* and *Placing towards arm*. *Picking up* is same as described in Section 4.2. *Placing towards arm* learns to move a small block (half size of the block used in the composite task) in the hand towards the robotic arm and then place it on the table. The agent fails if it cannot place the block to the target location.

**Composite task.** The JACO BAR-MOVING task requires the two arms to work very closely together. For example, the *Picking up* skill of both arms should be synchronized when they start to lift the bar and two arms require to lift the bar while maintaining the relative position between them since they are connected by holding the bar. The modular framework without explicit coordination of skills (Modular) can synchronize the execution of picking, moving, and placing. But the inability to micro-adjust the movement of the other arm causes instability of bar picking and moving. This results in degraded success rates compared to the modular framework with explicit coordination. Meanwhile, all baselines without pre-defined primitive skills fail to learn JACO BAR-MOVING.

### 4.4 ANT PUSH

We developed a multi-ant environment, ANT-PUSH, inspired from Nachum et al. (2019), simulated in the MuJoCo (Todorov et al., 2012) physics engine. We use the ant model in OpenAI Gym (Brockman et al., 2016). In this environment, two ants need to push a large object toward a green target place, collaborating with each other to keep the angle of the object as stable as possible (see Figure 4c).

**Primitives skills.** We train walking skills of an ant agent in 4 directions: up, down, left, and right. During primitive skill training, a block (half size of the block used in the composite task) and an ant agent are randomly placed. Pushing the block gives an additional reward to the agent, which prevents an ant to avoid the block. The learned primitive skills have different speed and trajectories conditioned on the latent behavior embedding.

**Composite task.** Our method achieves 32.3% success rate on ANT PUSH task while all baselines fail to compose primitive skills as shown in Figure 5c and Table 1. The poor performance of RL, MARL, RL-SBD, and MARL-SBD baselines shows the difficulty of credit assignment between agents, which leads one of the ants moves toward a block and pushes it but another ant does not move. Moreover, the Modular baseline with primitive skills also fails to learn the pushing task. This result illustrates the importance of coordination of agents, which helps synchronizing and controlling the velocities of both ant agents to push the block toward the goal position while maintaining its rotation.

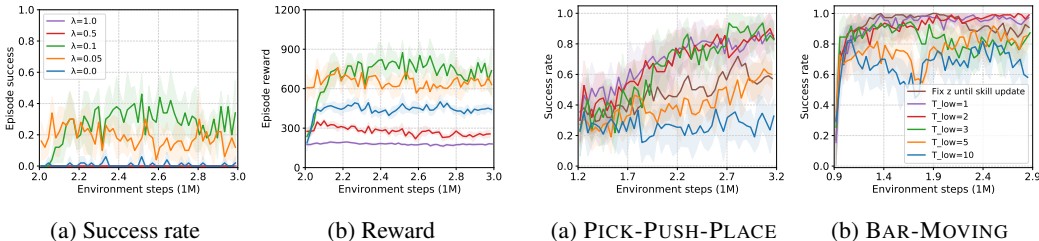

(a) Success rate    (b) Reward    (a) PICK-PUSH-PLACE    (b) BAR-MOVING

Figure 6: Learning curves of our method with different diversity coefficients $\lambda_2$ on ANT PUSH.

Figure 7: Success rates of our method with different $T_{low}$ coefficients on Jaco environments.

### 4.5 EFFECT OF DIVERSITY OF PRIMITIVE SKILLS

To analyze the effect of the diversity of primitive skills, we compare our model with primitive skills trained with different diversity coefficients $\lambda_2 = \{0.0, 0.05, 0.1, 0.5, 1.0\}$ in Equation (4) on ANT PUSH. Figure 6 shows that with small diversity coefficients $\lambda_2 = \{0.05, 0.1\}$, the agent can control detailed behaviors of primitive skills while primitive skills without diversity ($\lambda_2 = 0$) cannot be coordinated. The meta policy tries to synchronize two ant agents' positions and velocities by switching primitive skills, but it cannot achieve proper coordination without diversified skills. On the other hand, large diversity coefficients $\lambda_2 = \{0.5, 1.0\}$ make the primitive skills often focus on demonstrating diverse behaviors and fail to achieve the goals of the skills. Hence, these primitive skills do not have enough functionality to solve the target task. The diversity coefficient needs to be carefully chosen to acquire primitive skills with good performance as well as diverse behaviors.

### 4.6 EFFECT OF SKILL SELECTION INTERVAL $T_{low}$

To analyze the effect of the skill selection interval hyperparameter $T_{low}$, we compare our method trained with $T_{low} = \{1, 2, 3, 5, 10\}$ on Jaco environments. The success rate curves in Figure 7 demonstrate that smaller $T_{low}$ values in range $[1, 3]$ lead to better performance. This can be because the agent can realize more flexible skill coordination by adjusting the behavior embedding frequently.

In addition to the fixed $T_{low}$ values, we also consider the variation of our method in which the skill behavior embedding is only sampled when the meta policy updates its skill selection. Concretely, we set the value of $T_{low}$ to 1 but update $(z_t^1, \ldots, z_t^N)$ only if $(c_t^1, \ldots, c_t^N) \neq (c_{t-1}^1, \ldots, c_{t-1}^N)$. We observe that in this setting, the meta policy at times switch back and forth between two skills in two consecutive time steps, leading to slightly worse performance compared to our method with small $T_{low}$ values. This indicates that the meta policy needs to adjust the behavior embedding in order to optimally coordinate skills of the different agents.

## 5 CONCLUSION

In this paper, we propose a modular framework with skill coordination to tackle challenges of composition of sub-skills with multiple agents. Specifically, we use entropy maximization with mutual information maximization to train controllable primitive skills with diverse behaviors. To coordinate learned primitive skills, the meta policy predicts not only the skill to execute for each agent (end-effector) but also the behavior embedding that controls the chosen primitive skill's behavior. The experimental results on robotic manipulation and locomotion tasks demonstrate that the proposed framework is able to efficiently learn primitive skills with diverse behaviors and coordinate multiple agents (end-effectors) to solve challenging cooperative control tasks. Acquiring skills without supervision and extending our method to a visual domain are exciting directions for future work.

### ACKNOWLEDGMENTS

This project was funded by SKT. The authors would like to thank Karl Pertsch and many members of the USC CLVR lab for helpful discussion.

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

## A   Environment details

The details of observation spaces, action spaces, number of agents, and episode lengths are described in Table 2. All units in this section are in meters unless otherwise specified.

|  | Jaco Pick-Push-Place | Jaco Bar-Moving | Ant Push |
|---|---|---|---|
| Observation Space | 88 | 88 | 100 |
| - Robot observation | 62 | 62 | 82 |
| - Object observation | 26 | 26 | 18 |
| Action Space | 18 | 18 | 16 |
| Number of Agents | 2 | 2 | 2 |
| Episode length | 150 | 100 | 200 |

Table 2: Environment details

### A.1   Environment descriptions

In both Jaco environments, the robot works on a table with size $(1.6, 1.6)$ and top center position $(0, 0, 0.82)$. The two Jaco arms are initialized at positions $(-0.16, -0.16, 1.2)$ and $(-0.16, 0.24, 1.2)$. Left arm and right arm objects are initialized around $(0.3, 0.2, 0.86)$ and $(0.3, -0.2, 0.86)$ respectively in all primitive training and composite task training environments, with small random position and rotation perturbation.

In the *Jaco Pick-Push-Place* task, the right jaco arm needs to pick up the object and place it into the container initialized at the other side of the table. Success is defined by contact between the object and the inner top side of the container.

In the *Jaco Bar-Moving* task, the two Jaco arms need to together pick the long bar up by height of $0.7$, move it towards the arms by distance of $0.15$, and place it back on the table. Success is defined by (1) the bar being placed within $0.04$ away from the desired destination both in height and in xy-position and (2) the bar having been picked $0.7$ above the table.

In the *Ant Push* task, the two ant agents need to push a big box together to the goal position. The box has a size of $8.0 \times 1.6 \times 1.6$. The distance between ants and the box is $20$ cm and the distance between the box and the goal is $30$ cm. Initial positions have $1$ cm of randomness and the agent has a randomness of $0.01$ in each joint. The task is considered as success when both the distances between left and right end of the box and the goal are within $5$ cm.

### A.2   Reward design

For every task, we add a control penalty, $-0.001 * \|a\|^2$, to regularize the magnitude of actions where $a$ is a torque action performed by an agent.

**Jaco Pick:** To help the agent learn to reach, pick, and hold the picked object, we provide dense reward to the agent defined by the weighted sum of pick reward, gripper-to-cube distance reward, cube position and quaternion stability reward, hold duration reward, success reward, and robot control reward. More concretely,

$$R(s) = \lambda_{pick} \cdot (z_{box} - z_{init}) + \lambda_{dist} \cdot \text{dist}(p_{gripper}, p_{box}) + \lambda_{pos} \cdot \text{dist}(p_{box}, p_{init}) +$$
$$\lambda_{quat} \cdot \text{abs}(\Delta_{quat}) + \lambda_{hold} \cdot t_{hold} + \lambda_{success} \cdot \mathbf{1}_{\text{success}} + \lambda_{ctrl} \|a\|^2,$$

where $\lambda_{pick} = 500, \lambda_{dist} = 100, \lambda_{pos} = 1000, \lambda_{quat} = 1000, \lambda_{hold} = 10, \lambda_{success} = 100, \lambda_{ctrl} = 1 \times 10^{-4}$.

**Jaco Place:** Reward for place primitive is defined by the weighted sum of xy-distance reward, height reward (larger when cube close to floor), success reward, and robot control reward.

$$R(s) = \lambda_{xy} \cdot \text{dist}_{xy}(p_{box}, p_{goal}) + \lambda_z \cdot |z_{box} - z_{goal}| + \lambda_{success} \cdot \mathbf{1}_{\text{success}} + \lambda_{ctrl} \|a\|^2,$$

where $\lambda_{xy} = 500, \lambda_z = 500, \lambda_{success} = 500, \lambda_{ctrl} = 1 \times 10^{-4}$.

**Jaco Push:** Reward for push primitive is defined by the weighted sum of gripper reaching reward, box-to-destination distance reward, quaternion stability reward, hold duration reward, success reward, and robot control reward.

$$R(s) = \lambda_{reaching} \cdot \text{dist}(p_{gripper}, p_{box}) + \lambda_{pos} \cdot \text{dist}(p_{box}, p_{dest}) +$$
$$\lambda_{quat} \cdot \text{abs}(\Delta_{quat}) + \lambda_{hold} \cdot t_{hold} + \lambda_{success} \cdot \mathbf{1}_{\text{success}} + \lambda_{ctrl} \|a\|^2,$$

where $\lambda_{reaching} = 100, \lambda_{pos} = 500, \lambda_{quat} = 30, \lambda_{hold} = 10, \lambda_{success} = 1000, \lambda_{ctrl} = 1 \times 10^{-4}$.

**Jaco Pick-Push-Place:** Reward for Pick-Push-Place is defined by the weighted sum of gripper contact reward, per-stage reach/pick/push/place rewards, success reward, and control reward. We tune the reward carefully for all baselines.

$$R(s) = \lambda_{contact} \cdot \left(\mathbf{1}_{\text{left gripper touches container}} + \mathbf{1}_{\text{right gripper touches box}}\right) +$$
$$\lambda_{reach} \cdot \mathbf{1}_{\text{reach}} \cdot \left(\text{dist}(p_{left\_gripper}, p_{container}) + \text{dist}(p_{right\_gripper}, p_{box})\right) +$$
$$\lambda_{pick} \cdot \mathbf{1}_{\text{pick}} \cdot \text{dist}(p_{box}, p_{box\_target}) + \lambda_{place} \cdot \mathbf{1}_{\text{place}} \cdot \text{dist}(p_{box}, p_{box\_target}) +$$
$$\lambda_{push} \cdot \text{dist}(p_{container}, p_{container\_target}) + \lambda_{success} \cdot \mathbf{1}_{\text{success}} + \lambda_{ctrl} \cdot \|a\|^2,$$

where $\lambda_{reach} = 10, \lambda_{contact} = 10, \lambda_{pick} = \lambda_{place} = \lambda_{place} = 10, \lambda_{success} = 50, \lambda_{ctrl} = 0$, and $\mathbf{1}_{\text{reaching}}, \mathbf{1}_{\text{pick}}$, and $\mathbf{1}_{\text{place}}$ are indicator functions specifying whether the agent is in reaching, pick or place stage. Agent stages are determined by how many multiples of 25 steps the agent has stepped through in the environment.

**Jaco Bar-Moving:** Reward for Bar-Moving is defined by the weighted sum of per-stage reach/pick/move/place rewards, success reward, and control reward.

$$R(s) = \lambda_{reach} \cdot \mathbf{1}_{\text{reach}} \cdot \left(\text{dist}(p_{left\_gripper}, p_{left\_handle}) + \text{dist}(p_{right\_gripper}, p_{right\_handle})\right) +$$
$$\lambda_{pick} \cdot \mathbf{1}_{\text{pick}} \cdot \text{dist}(p_{bar}, p_{bar\_target}) + \lambda_{move} \cdot \mathbf{1}_{\text{place}} \cdot \text{dist}_{xy}(p_{bar}, p_{bar\_target}) +$$
$$\lambda_{place} \cdot \mathbf{1}_{\text{place}} \cdot \text{dist}_z(p_{bar}, p_{bar\_target}) + \lambda_{success} \cdot \mathbf{1}_{\text{success}} + \lambda_{ctrl} \cdot \|a\|^2,$$

where $\lambda_{reach} = 10, \lambda_{pick} = 30, \lambda_{move} = 100, \lambda_{place} = 100, \lambda_{success} = 100, \lambda_{ctrl} = 1 \times 10^{-4}$, and $\mathbf{1}_{\text{pick}}$ and $\mathbf{1}_{\text{place}}$ are indicator functions specifying whether the agent is in pick or place stage. Agent stages are determined by whether the pick objective is fulfilled or not.

**Ant Push & Ant Moving:** Reward for ANT PUSH is defined by upright, velocity towards the desired direction. We provide a dense reward to encourage the desired locomotion behavior using velocity, stability, and posture, as following:

$$R(s) = \lambda_{vel} \cdot \text{abs}(\Delta x_{ant}) + \lambda_{boxvel} \cdot \text{abs}(\Delta x_{box}) + \lambda_{upright} \cdot \cos(\theta) - \lambda_{height} \cdot \text{abs}(0.6 - h) +$$
$$\lambda_{goal} \cdot \text{dist}(p_{goal}, p_{box}),$$

where $\lambda_{vel} = 50, \lambda_{boxvel} = 20, \lambda_{upright} = 1, \lambda_{height} = 0.5$. For ANT PUSH, we provide an additional reward based on distance between the box and the goal position with $\lambda_{goal} = 200$.

# B  EXPERIMENT DETAILS

We use PyTorch (Paszke et al., 2017) for our implementation and all experiments are conducted on a workstation with Intel Xeon Gold 6154 CPU and 4 NVIDIA GeForce RTX 2080 Ti GPUs.

## B.1 HYPERPARAMETERS

| Parameters | Value |
|---|---|
| learning rate | 3e-4 |
| gradient steps | 50 |
| batch size | 256 |
| discount factor | 0.99 |
| target smoothing coefficient | 0.005 |
| reward scale (SAC) | 1.0 |
| experience buffer size (# episodes) | 1000 |
| $T_{\text{low}}$ | 1 for JACO, 5 for ANT |
| $N_z$ (dimensionality of $z$) | 5 |

Table 3: Hyperparameters

## B.2 NETWORK ARCHITECTURES

**Actor Networks:** In all experiments, we model our actor network for each primitive skill as a 3-layer MLP with hidden layer size 64. The last layer of the MLP is two-headed – one for the mean of the action distribution and the other for the standard deviation of it. We use ReLU as activation function in hidden layers. We do not apply any activation function for the final output layer. The action distribution output represents per-dimension normal distribution, from which single actions can be sampled and executed in the environment.

**Critic Networks:** The critic network for each primitive skill and meta policy is modeled as a 2-layer MLP with hidden layer size 128. ReLU is used as an activation function in the hidden layers. The critic network output is used to assess the value of a given state-action pair, and is trained by fitting its outputs to the target Q-value clamped by $\pm 100$.

**Meta Policy:** The meta policy is modeled as a 3-layer MLP with hidden layer size of 64. Since the actions of meta policy are sampled from $N$ categorical distributions for each end-effector/agent and $N$ normal distributions for behavior embeddings, the output dimension of the meta policy is $\sum_{i=1}^{N}(m^i + N_z)$. The meta policy uses ReLU as an activation function for all layers except for the final output layer.

## B.3 TRAINING DETAILS

For all baselines, we train the meta policies using PPO and the low-level policies using SAC. We use the same environment configurations, composite task reward definitions, and value of $T_{low}$ across all baselines.

For Jaco tasks, we train a total of 4 primitive skills – right arm pick, right arm place-to-center, right arm place-towards-arm, and left arm push – to be composed by meta-policy. For Jaco Pick-Push-Place, we provide the meta-policy with right arm pick and right arm place-to-center as right arm primitives and left arm push as left arm primitives; for Jaco Bar-Moving, we provide the meta-policy with right arm pick and right arm place-towards-arm as both right and left arm primitives and left arm pick and right arm place-towards-arm as left arm primitives. We obtain left arm primitives for bar-moving task by using the learned right arm primitives directly.

To obtain the 4 primitives skills described above, we train right arm pick with diversity coefficient $\lambda_2 = 0.01$ and the other three primitives with $\lambda_2 = 0.1$. The destination of right arm push is set to $(0.3, -0.03, 0.86)$, which is slightly left of the center of the table. After pick primitive is trained, we train the two right arm place primitives where episodes are initialized by intermediate states of successful right arm pick episodes where the height of the box is larger than 0.94 (0.01 higher than the target pick height). The place destinations for towards-arm and to-center primitives are $(0.15, -0.2, 0.86)$ and $(0.3, -0.02, 0.86)$, respectively.

For non-modular baselines that incorporates skill behavior diversification, we use $\lambda_2 = 0.01$ for both Jaco Pick-Push-Place and Jaco Bar-Moving because both tasks require picking skills, which can only be trained with a small value of $\lambda_2$.

