# OpenReview forum: "Learning to Coordinate Manipulation Skills via Skill Behavior Diversification"
_ICLR.cc/2020/Conference — Accept (Poster)_

### Official Review · AnonReviewer2 · 2019-10-22
**Official Blind Review #2**

**Rating:** 6

**Review:**

This paper aims to achieve multi-agent coordination by composing diverse skills learned by augmenting individual subtask objectives with DIAYN-style diversity bonuses. Once individual diverse skills are learned for the subtasks, the agents are combined by a meta-agent to coordinate multiple distinct robots to achieve a shared goal.

This is a good application of low-level skill learning to multi-agent coordination. I have settled on a weak acceptance, because the approach is simple and seems scalable, but the acceptance is weak because the method relies on specifying the subtasks in advance.

The approach is well-motivated in that learning individual skills in isolation is generally more tractable than learning their combined application from scratch, and the building blocks of this system are well-chosen. The results demonstrate the importance of the diversity objective, and find a good sweet spot for the diversity weight.

I do have some criticisms, related primarily to the decision to pre-train with both a continuously-parameterized diversity conditioning as well as a discrete set of concrete subtasks. Because these subtasks must be specified in advance, this limits the wide applicability of the resulting approach to those that can be broken down a-priori into components. Did the authors consider using the DIAYN objective on its own to encourage sufficiently diverse behaviors? If this didn't work, would perhaps a larger latent skill vector, or a large discrete set of DIAYN skills, have made it work?

I also don't see the size of the latent skill embedding reported anywhere. How big is this vector; that information should be added to the paper.

However, the approach is generally good. I think the paper would be improved if it included a strong baseline that uses DIAYN only (no a-priori subtasks), so we can evaluate how important that expert knowledge is to the final performance.

**Experience Assessment:**

I have read many papers in this area.

**Review Assessment: Checking Correctness Of Derivations And Theory:**

I assessed the sensibility of the derivations and theory.

**Review Assessment: Checking Correctness Of Experiments:**

I assessed the sensibility of the experiments.

**Review Assessment: Thoroughness In Paper Reading:**

I read the paper thoroughly.

---

> ### Author Response · Authors · 2019-11-15
> **Response to Reviewer 2**
>
> We thank the reviewer for the constructive feedback and address the concerns in detail below.
>
>
> - Subtasks must be specified in advance. … Did the authors consider using the DIAYN objective on its own to encourage sufficiently diverse behaviors? If this didn't work, would perhaps a larger latent skill vector, or a large discrete set of DIAYN skills, have made it work?
>
> Thank you for suggesting an interesting idea! Solving complex tasks using DIAYN skills [1] conditioned on a large latent skill vector without a priori knowledge is an interesting direction. If DIAYN skills without a priori subtasks have sufficiently diverse behaviors, our method can coordinate those skills by selecting the latent skill vector. However, we feel that it is out of the scope of this project.
>
> As Reviewer 2 (R2) noted in “learning individual skills in isolation is generally more tractable than learning their combined application from scratch”, the focus of this paper is to independently learn flexible skills that can be composed with other skills of other end-effectors/agents, and to coordinate those skills to solve more complicated and cooperative tasks.
>
> Please note that our method is complementary to the suggestion R2 pointed out. As R2 suggested, unsupervisedly acquiring skills for more complicated tasks is an important problem yet orthogonal and complementary to our method. We will leave this problem as a future work of the community.
>
>
> - The paper would be improved if it included a strong baseline that uses DIAYN only (no a priori subtasks), so we can evaluate how important that expert knowledge is to the final performance.
>
> The tasks discussed in this paper deal with coordination between complex primitive skills of several agents or agent body parts. We found that the baseline R2 suggested cannot solve these tasks mainly because DIAYN skills [1] acquired without a priori subtasks do not capture complicated interactions with the environment required for the final tasks. This result shows that prior knowledge about subtasks takes a critical role in tackling complex and cooperative tasks.
>
> To show this concretely, we have added two baselines (Section 4.1) — centralized policy and decentralized policy with Skill Behavior Diversification (centralized SBD and decentralized SBD), as suggested by R2. The centralized SBD baseline has a single low-level policy trained with DIAYN while the decentralized SBD baseline has multiple low-level policies trained with DIAYN. Both baselines have a meta policy, which controls low-level policies by generating behavior embeddings, and do not have an external skill-specific reward.
>
> These baselines have a hard time learning useful skills for skill coordination towards the final composite task, which indicates domain expert knowledge becomes more critical as tasks become more complicated. On the other hand, our proposed method can successfully solve the final tasks with pre-trained primitive skills. We have added these results in Figure 4 and Table 1.
>
>
> - The size of latent skill embedding is not reported.
>
> We thank R2 for pointing out missing information about the size of latent skill embedding. We used 5 for the size of latent behavior embedding. We have added this information in Section 3.5 and supplementary material (Appendix C.1).
>
>
> [1] Eysenbach et al. “Diversity is All You Need: Learning Skills without a Reward Function”, ICLR 2019

---

### Official Review · AnonReviewer3 · 2019-10-23
**Official Blind Review #3**

**Rating:** 6

**Review:**

This paper provides a specific way of incorporating temporal abstraction into the multi-agent reinforcement learning (MARL) setting. Specifically, this method first discovers diversified skills for every single agent and then train a meta-policy to choose among skills for all agents.

Overall, this paper is well presented so I can understand it well. Unfortunately, this paper didn't give me too much scientific insight. As maybe this is because I don't know too much about MARL, I would like to ask the author to help me address the following questions.

My first key question is, should we treat temporal abstraction (TA) under the multi-agent setting different from it under the single-agent setting? If they are the same, why do we bother discussing TA under the multi-agent setting? Why not just discuss it under the simpler single-agent setting? If they are not, what are the differences?

The second key question is if TA under the multi-agent system is special, then why the DIAYN method, which is proposed under the single-agent setting, could be directly used in the multi-agent setting? Why should we consider the DIAYN method, instead of other skills discovery methods?

Furthermore, I would also like the author to help me address three more concrete questions.

1. Section 1, paragraph 2, the author wrote: "However, all these approaches are focused on working with a single end-effector or agent with learned primitive skills, and learning to coordinate has not been addressed." Does the author mean there is no temporal abstraction method for multiple collaborative agents?

2. Section 3.3, the author wrote, " the prior distribution p(z) is Gaussian." I.e., Z is continuous r.v. I would like to know how the author could learn q(z|s) to approximate p(z|s), which is an arbitrary continuous distribution. Maybe I am wrong, but I don't see a way to do this.

3. In algorithm 1, a skill, once being chosen, will be executed for T_{low} steps, where T_{low} is fixed and pre-defined by the algorithm designer. I would like to hear to author analyzing the pros and cons of this critical design choice.

**Experience Assessment:**

I do not know much about this area.

**Review Assessment: Checking Correctness Of Derivations And Theory:**

I assessed the sensibility of the derivations and theory.

**Review Assessment: Checking Correctness Of Experiments:**

I assessed the sensibility of the experiments.

**Review Assessment: Thoroughness In Paper Reading:**

I read the paper at least twice and used my best judgement in assessing the paper.

---

> ### Author Response · Authors · 2019-11-15
> **Response to Reviewer 3**
>
> We thank the reviewer for the constructive feedback and address the concerns in detail below.
>
>
> - Should we treat temporal abstraction under the multi-agent setting different from it under the single-agent setting?
>
> We treat temporal abstraction (TA) under the multi-agent system the same as TA under the single-agent setting. However, utilizing temporally-extended actions (primitive skills) in the multi-agent system introduces the coordination problem between agents where skills from different agents disturb each other or are not able to complete collaborative tasks. Please note that this paper focuses on tackling this problem by learning to coordinate skills.
>
>
> - Why should we consider the DIAYN method, instead of other skills discovery methods?
>
> Diverse skills discovered by any skill discovery methods can be used by our method as long as acquired skills are controllable and diverse enough to complete the final task. In this paper, we choose DIAYN for discovering diverse behaviors and corresponding latent representations given a reward function.
>
>
> - “However, all these approaches are focused on working with a single end-effector or agent with learned primitive skills, and learning to coordinate has not been addressed.” Does the author mean there is no temporal abstraction method for multiple collaborative agents?
>
> The sentence R3 mentioned refers to the modular approaches which suggest to learn reusable skills and combine them to solve more complex tasks. As far as we know, there is no prior work about tackling the coordination problem between multiple agents with skills learned in isolation.
>
> There are a few works [2-4] utilizing temporal abstraction for multi-agent RL. However, the applications of these methods are limited to simple 2d tasks since all skills need to be learned from scratch. Instead, our method utilizes primitive skills to solve complex tasks and tackles the coordination problem which happens when multiple agents perform their skills without considering each other.
>
>
> - Analyze the design choice “a skill, once being chosen, will be executed for $T_{low}$ steps, where $T_{low}$ is fixed and pre-defined by the algorithm designer”.
>
> A fixed horizon $T_{low}$ has been actively used in HRL [11-14] due to its simplicity and efficient implementation. Instead of using a fixed $T_{low}$, skill termination function [6-8] can be learned to support variable lengths of primitive skills and flexible skill selection. There are also works [9,10] that vary the horizon for the low-level policy with the goal specified by the meta policy. However, joint learning of which skill to select and when to switch skills becomes difficult for the meta policy due to the credit assignment problem.
>
> We let $T_{low}$ as a tunable hyperparameter according to tasks. If primitive skills have a long horizon and not sensitive to temporal alignment, larger $T_{low}$ can provide faster learning by utilizing temporal abstraction. On the other hand, if the tasks require sharp changes over primitive skills, small $T_{low}$ is more suitable. Additional experiments (Figure 7) show that the agent can realize more flexible skill coordination with small $T_{low}$ by adjusting the behavior embedding more frequently for our tasks, while switches skills only when required.
>
>
> - In section 3.3, how does the method learn $q(z|s)$ to approximate $p(z|s)$?
>
> We learn $q(z|s)$ to approximate $p(z|s)$ by maximizing the reward (Equation 4) that includes the variational lower bound [1,5]. Maximizing the variational lower bound minimizes KL-divergence between $q(z|s)$ and $p(z|s)$, which means $q(z|s)$ approximates $p(z|s)$. Please refer to [1] for more details.
>
>
>
> [1] Eysenbach et al. “Diversity is All You Need: Learning Skills without a Reward Function”, ICLR 2019
> [2] Tang et al. “Hierarchical Deep Multiagent Reinforcement Learning with Temporal Abstraction”, arXiv 2018
> [3] Han et al. “Multi-Agent Hierarchical Reinforcement Learning with Dynamic Termination”, AAMAS Extended Abstract 2019
> [4] Ahilan et al. “Feudal Multi-Agent Hierarchies for Cooperative Reinforcement Learning”, arXiv 2019
> [5] Kingma et al. “Auto-Encoding Variational Bayes”, ICLR 2014
> [6] Bacon et al. “The Option-Critic Architecture”, AAAI 2017
> [7] Andreas et al. “Modular Multitask Reinforcement Learning with Policy Sketches”, ICML 2017
> [8] Oh et al. “Zero-shot Task Generalization with Multi-Task Deep Reinforcement Learning”, ICML 2017
> [9] Levy et al. “Hierarchical Reinforcement Learning with Hindsight”, ICLR 2019
> [10] Lee et al. “To Follow or not to Follow: Selective Imitation Learning from Observations”, CoRL 2019
> [11] Frans et al. “Meta Learning Shared Hierarchies”, ICLR 2018
> [12] Co-Reyes et al. “Self-consistent trajectory autoencoder: Hierarchical reinforcement learning with trajectory embeddings”, ICML 2018
> [13] Nachum et al. “Data-efficient hierarchical reinforcement learning.” NeurlPS 2018
> [14] Lee et al. “Composing Complex Skills by Learning Transition Policies”, ICLR 2019

---

### Official Review · AnonReviewer1 · 2019-10-24
**Official Blind Review #1**

**Rating:** 6

**Review:**

The paper presents a hierarchical reinforcement learning method for coordination of multiple cooperative agents with pre-learned adaptable skills. These skills are learned via a maximum entropy objective where diversity of behaviour given each skill is maximised and controlled via a latent conditioning vector. This allows controllability of the variation in skill execution by the meta-policy via changing the skill-specific latent vectors. The paper presents empirical results in manipulation (pick-push-place and moving a long bar by coordinating two Jaco arms) and locomotion (two Ants pushing a large block to a goal location). The method proposed outperforms the baselines reported.

Overall, this paper addresses an interesting problem and can be impactful with the caveat for some clarifications and analysis. Given that the authors address my concerns, I would be willing to increase my score.

The main novelty of this work lies in how one can learn sub-skills that can be leveraged and adapted for down-stream tasks. The problem setting used to test the method is a multi-agent setting where it is crucial that skills are adapted to enable cooperation. I found the environments and the problem setting generally interesting and important for testing the proposed method. I have a few concerns that I listed below:


1) I found the notations at times inconsistent and confusing. It would have helped to see some more details on the diagram (Figure 2) to understand how everything fits together.

2) The set of skills for the two agents are selected by the meta-policy in every T_low steps. It looks like in the Jaco environments T_low = 1. Can you comment on this? This seems slightly concerning since it seems like the meta-controller is treating these skills as primitive actions rather than temporally extended behaviour.

3) Looking at the training curves in Figure 4, there seems to be a really high variance in performance of the method. Can you comment on this as this seems concerning. Could you run more seeds to improve this?

4) It is nice to see in section 4.5 how the hyper-parameters balancing diversity in combination with external reward (equation 4) is tuned and how sensitive that is to achieving adaptability for downstream tasks. The only criticism I have is that it is difficult to understand from "Episode reward" on y-axis what the success rate is (similar to Figure 4)? It would’ve been nice to report results in a consistent way throughout the paper for these environments.

5) Given that all the tasks in the experiments are cooperative multi-agent settings, I would have liked to see more in depth discussion regarding alternative multi-agent methods. The multi-agent baseline provided (which is using a decentralized policy with a shared critic, inspired by Lowe et al., 2017) seems fair, but I wonder if there has been more recent work in this direction that could have been highlighted?



**Experience Assessment:**

I have read many papers in this area.

**Review Assessment: Checking Correctness Of Derivations And Theory:**

N/A

**Review Assessment: Checking Correctness Of Experiments:**

I carefully checked the experiments.

**Review Assessment: Thoroughness In Paper Reading:**

I read the paper at least twice and used my best judgement in assessing the paper.

---

> ### Author Response · Authors · 2019-11-15
> **Response to Reviewer 1**
>
> We thank the reviewer for the constructive feedback and address the concerns in detail below.
>
>
> - The meta-controller is treating skills as primitive actions rather than temporally extended behavior with $T_{low}=1$.
>
> As Reviewer 1 (R1) mentioned, with $T_{low}=1$, the meta policy can treat primitive skills as primitive actions rather than temporally extended behaviors. In this paper, we focus on behavioral coordination of skills rather than exploiting temporal abstraction of skills, and we found that adjusting the behavior of skills frequently is critical to the performance of the final tasks.
>
> To investigate the effect of skill selection interval $T_{low}$, we conducted an additional experiment on the Jaco environments with different $T_{low}$ values (1, 2, 3, 5, 10) and added the results in Figure 7 (Appendix A.2). The results show that smaller $T_{low}$ values (1, 2, 3) perform better than larger $T_{low}$ values (5, 10). This can be because the agent can realize more flexible skill coordination by adjusting the behavior embedding more frequently.
>
> To verify our reasoning above, we came up with a variation of our method in which the meta policy can choose a primitive skill to execute every time step but choose a behavior embedding only when a primitive skill changes. In this setting, the meta policy at times switches back and forth between two skills in consecutive time steps to change the behavior embedding. This results in worse performance compared to our method with a small $T_{low}$ which switches skills only when required, but adapts behavior embeddings frequently. This indicates that the meta policy needs to frequently adjust the behavior embedding in order to optimally coordinate the skills of the different agents.
>
>
> - More in-depth discussion regarding alternative multi-agent methods.
>
> To the best of our knowledge, there is no prior work about tackling the coordination problem between multiple agents with skills learned in isolation. Existing multi-agent reinforcement learning methods either learn communication mechanisms [1-3] or attempt to resolve the credit assignment problem during training [4-7] between agents. However, communication mechanisms cannot be utilized when primitive skills are pre-trained separately without communication. Also, credit assignment problems become more challenging to resolve when the complexity of cooperative tasks increases and all agents need to learn completely from scratch.
>
> There are a few works [8-10] utilizing multiple layers of hierarchies for multi-agent RL. However, the applications of these methods are limited to tasks in simple 2d environments given that all skills need to be learned from scratch. Instead, we propose to utilize primitive skills to solve complex tasks and tackle the coordination problem which happens when multiple agents perform their skills without considering other agents.
>
>
> - High variance in performance of the method.
>
> As R1 pointed out, Figure 4 shows a high variance since the curves are plotted without moving average. We updated all curves (Figure 4-7) with moving average of 10 evaluations. Moreover, we ran experiments with more seeds, so all Jaco experiments are reported using 6 different random seeds.
>
>
> - Analysis plot in Section 4.5 shows episode rewards but not success rates.
>
> We choose the episode reward over the success rate since most experiments fail to succeed on the task, which makes the differences between experiments less noticeable. As suggested by R1, we have also added a plot of success rates in Figure 6 in the supplementary material (Appendix A.1).
>
>
> - Notations at times inconsistent and confusing.
>
> We thank for pointing out inconsistent notations and suggesting an idea of displaying notations in Figure 2. We revised the paper for consistent notations and Figure 2 to provide a better understanding of our framework.
>
>
> [1] Sukhbaatar et al. “Learning Multiagent Communication with Backpropagation”, NIPS 2016
> [2] Peng et al. “Multiagent bidirectionally-coordinated nets: Emergence of human-level coordination in learning to play starcraft combat games”, arXiv 2017
> [3] Jiang et al. “Learning Attentional Communication for Multi-Agent Cooperation”, NIPS 2018
> [4] Tan et al. “Multi-agent reinforcement learning: Independent vs. cooperative agents,” ICML 1993
> [5] Sunehag et al. “Value-Decomposition Networks For Cooperative Multi-Agent Learning”, AAMAS 2018
> [6] Rashid et al. “QMIX: Monotonic Value Function Factorisation for Deep Multi-Agent Reinforcement Learning”, ICML 2018
> [7] Foerster et al. “Counterfactual Multi-Agent Policy Gradients”, AAAI 2018
> [8] Tang et al. “Hierarchical Deep Multiagent Reinforcement Learning with Temporal Abstraction”, arXiv 2018
> [9] Han et al. “Multi-Agent Hierarchical Reinforcement Learning with Dynamic Termination”, AAMAS Extended Abstract 2019
> [10] Ahilan et al. “Feudal Multi-Agent Hierarchies for Cooperative Reinforcement Learning”, arXiv 2019

---

### Author Response · Authors · 2019-11-15
**Response to Reviewers**

We thank all the reviewers for providing constructive feedback and have updated our paper accordingly. We first summarize concerns from reviewers here and then respond to individual reviews.


- DIAYN only baseline without a priori subtasks to help evaluate the importance of expert knowledge (predefined subtasks) to the final performance. [R2]

The tasks in this paper deal with coordination between complex skills of multiple agents. As shown in Figure 4 and Table 1, we found that the baseline suggested by R2 fails to solve these tasks because DIAYN skills [1] acquired without a priori subtasks do not capture complicated interactions with the environment required for the final tasks. This result shows that prior knowledge about subtasks takes a critical role in tackling complex tasks.

To show this concretely, we have added two baselines: centralized policy and decentralized policy with Skill Behavior Diversification (centralized SBD and decentralized SBD). The centralized SBD baseline has a single low-level policy trained with DIAYN while the decentralized SBD baseline has multiple low-level policies trained with DIAYN. Both baselines have a meta policy, which controls low-level policies by generating behavior embeddings, and do not have an external skill-specific reward. Both baselines fail to learn useful skills for the final tasks, while our method can successfully solve the tasks with pre-trained primitive skills. This shows that domain expert knowledge becomes more critical as tasks become more complicated.


- Can this method work with other skill discovery methods? [R2, R3]

Diverse skills discovered by any skill discovery method can be used by our method as long as acquired skills are controllable and diverse enough to complete the final task. Please note that our method is orthogonal to skill discovery methods.


- Design choice about the fixed horizon of the low-level skills $T_{low}$. [R1, R3]

A fixed horizon $T_{low}$ has been actively used in HRL [2-5] due to its simplicity and efficient implementation. Instead of using a fixed $T_{low}$, skill termination function [6-8] can be learned to support variable lengths of primitive skills and flexible skill selection. There are also works [9,10] that vary the horizon for the low-level policy with the goal specified by the meta policy. However, joint learning of which skill to select and when to switch skills becomes difficult for the meta policy due to the credit assignment problem.


- The meta-controller is treating skills as primitive actions rather than temporally extended behavior with $T_{low}=1$. [R1]

In this paper, we focus on behavioral coordination of skills rather than exploiting temporal abstraction of skills, and we found that adjusting the behavior of skills frequently is critical to the performance of the final tasks.

We let $T_{low}$ as a tunable hyperparameter according to tasks. If primitive skills have a long horizon and not sensitive to temporal alignment, larger $T_{low}$ can provide faster learning by utilizing temporal abstraction. On the other hand, if the task requires sharp changes over primitive skills, small $T_{low}$ is more suitable. Additional experiments (Figure 7) show that the agent can realize more flexible skill coordination with small $T_{low}$ by adjusting the behavior embedding more frequently for our tasks, while switches skills only when required.


- More discussion regarding related multi-agent methods. [R1, R3]

To the best of our knowledge, there is no prior work about tackling the coordination problem between multiple agents with skills learned in isolation. Existing multi-agent reinforcement learning methods either learn communication mechanisms [11-13] or attempt to resolve the credit assignment problem [14-17] between agents. However, communication mechanisms cannot be utilized when primitive skills are pre-trained separately without communication. Also, credit assignment problems become more challenging when the complexity of cooperative tasks increases and all agents need to learn completely from scratch. There are a few works [18-20] utilizing hierarchies for multi-agent RL. However, the applications of these methods are limited to simple 2d tasks since all skills need to be learned from scratch.

---

> ### Author Response · Authors · 2019-11-15
> **References**
>
>
> [1] Eysenbach et al. “Diversity is All You Need: Learning Skills without a Reward Function”, ICLR 2019
> [2] Frans et al. “Meta Learning Shared Hierarchies”, ICLR 2018
> [3] Co-Reyes et al. “Self-consistent trajectory autoencoder: Hierarchical reinforcement learning with trajectory embeddings”, ICML 2018
> [4] Nachum et al. “Data-efficient hierarchical reinforcement learning.” NeurlPS 2018
> [5] Lee et al. “Composing Complex Skills by Learning Transition Policies”, ICLR 2019
> [6] Bacon et al. “The Option-Critic Architecture”, AAAI 2017
> [7] Andreas et al. “Modular Multitask Reinforcement Learning with Policy Sketches”, ICML 2017
> [8] Oh et al. “Zero-shot Task Generalization with Multi-Task Deep Reinforcement Learning”, ICML 2017
> [9] Levy et al. “Hierarchical Reinforcement Learning with Hindsight”, ICLR 2019
> [10] Lee et al. “To Follow or not to Follow: Selective Imitation Learning from Observations”, CoRL 2019
> [11] Sukhbaatar et al. “Learning Multiagent Communication with Backpropagation”, NIPS 2016
> [12] Peng et al. “Multiagent bidirectionally-coordinated nets: Emergence of human-level coordination in learning to play starcraft combat games”, arXiv 2017
> [13] Jiang et al. “Learning Attentional Communication for Multi-Agent Cooperation”, NIPS 2018
> [14] Tan et al. “Multi-agent reinforcement learning: Independent vs. cooperative agents,” ICML 1993
> [15] Sunehag et al. “Value-Decomposition Networks For Cooperative Multi-Agent Learning”, AAMAS 2018
> [16] Rashid et al. “QMIX: Monotonic Value Function Factorisation for Deep Multi-Agent Reinforcement Learning”, ICML 2018
> [17] Foerster et al. “Counterfactual Multi-Agent Policy Gradients”, AAAI 2018
> [18] Tang et al. “Hierarchical Deep Multiagent Reinforcement Learning with Temporal Abstraction”, arXiv 2018
> [19] Han et al. “Multi-Agent Hierarchical Reinforcement Learning with Dynamic Termination”, AAMAS Extended Abstract 2019
> [20] Ahilan et al. “Feudal Multi-Agent Hierarchies for Cooperative Reinforcement Learning”, arXiv 2019

---

### Decision · Program_Chairs · 2019-12-19

**Decision:**

Accept (Poster)

**Comment:**

This paper deals with multi-agent hierarchical reinforcement learning. A discrete set of pre-specified low-level skills are modulated by a conditioning vector and trained in a fashion reminiscent of Diversity Is All You Need, and then combined via a meta-policy which coordinates multiple agents in pursuit of a goal. The idea is that fine control over primitive skills is beneficial for achieving coordinated high-level behaviour.

The paper improved considerably in its completeness and in the addition of baselines, notably DIAYN without discrete, mutually exclusive skills. Reviewers agreed that the problem is interesting and the method, despite involving a degree of hand-crafting, showed promise for informing future directions.

On the basis that this work addresses an interesting problem setting with a compelling set of experiments, I recommend acceptance.